# Big Five, Self-Regulation, and Coping Strategies as Predictors of Achievement Emotions in Undergraduate Students

**DOI:** 10.3390/ijerph17103602

**Published:** 2020-05-20

**Authors:** Jesús de la Fuente, Paola Paoloni, Douglas Kauffman, Meryem Yilmaz Soylu, Paul Sander, Lucía Zapata

**Affiliations:** 1School of Education and Psychology, University of Navarra, 31009 Pamplona, Spain; 2School of Psychology, University of Almería, 04120 Almería, Spain; 3National Scientific and Technical Research Council (CONICET), National University of Río Cuarto, Cordoba 5800, Argentina; paopaoloni17@hotmail.com; 4School of Medicine, Medical University of the Americas, Nevis, Devens, MS 01434, USA; douglas.f.kauffman@gmail.com; 5School of Psychology, University of Nebraska—Lincoln, Lincoln, NE 68588, USA; meryemy@gmail.com; 6School of Psychology, Teesside University, Middlesbrough TS1 3BA, UK; P.Sander@tees.ac.uk; 7Private Practice, Educational Psychologist, Cardiff CF62 5DL, UK; lucia.epsychology@gmail.com

**Keywords:** Big Five Model, self-regulation, coping strategies, achievement emotions, Structural Equation Model (SEM)

## Abstract

The study focused on the analysis of linear relations between personality, self-regulation, coping strategies and achievement emotions. The main objective was to establish a model of linear, empirical, associative to infer needs and proposals for intervening in emotional health in the different profiles of university students. A total of 642 undergraduate students participated in this research. Evidence of associative relations between personality factors, self-regulation and coping strategies was found. The neuroticism factor had a significant negative associative relationship with Self-Regulation both globally and in its factors; especially important was its negative relation to decision making, and coping strategies focused in emotion. The results of Structural Equation Model showed an acceptable model of relationships, in each emotional context. Results and practical implications are discussed.

## 1. Introduction

The behavioral study of the learning process in university students has traditionally focused on analyzing cognitive and metacognitive processes to explain optimal learning and achievement [1,2,3]. The prevailing psychological paradigm, however, has reoriented research interest toward the study of emotional processes, and the analysis of personality and emotional factors has become important [4,5]. For this reason, the present study has focused on the analysis of linear relations between different student variables, to date insufficiently examined by the dominant cognitivist paradigms, namely, the variables of personality, self-regulation, coping strategies and achievement emotions. 

### 1.1. Emotional Health at University

The emotional health needs of students and young people seem to be on the rise. At the global level, mental health problems will be one of the main causes of disability, morbidity and mortality within the next two decades [6]. Personality factors can influence one’s vulnerability to emotional problems and disorders, including anxiety, depression, aggression and personality dysfunction [7]. Recent research has put forward two types of psychological disorders. On one side, there are internalizing disorders (inhibition and negative affect), referring to depression, generalized anxiety, phobias, post-traumatic disorders and obsessive-compulsive disorders; these can be manifest in adults, children and adolescents. On the other side are externalizing disorders (aggression, behavior problems, lack of control over behavior), which are associated with temperament issues. In this category we find antisocial behavior, related to low scores in the traits of Agreeableness (A) or Conscientiousness (C), and high scores in Neuroticism (N). Personality traits have also been related to impulsivity, hostility and low self-esteem, which have been associated with different indices of aggression, delinquency and illegal behaviors [8]. Recent research has contributed plentiful evidence with respect to their positive or negative attitude at work [9] healthful behaviors in pregnant women [10]. Elsewhere, different personality types have been established, in relation to the Big Five, with pathogenic components. For example, perfectionism [11] has been defined with two elements, perfectionist concerns and perfectionist efforts [12], although it has also been differentiated from excellence and efficiency [13].

### 1.2. Big Five Personality Model in Educational Context of University

The psychological construction of personality refers to the relatively stable differences in behavior and in affective dispositions that are generalized to a broad range of environmental situations [14,15,16]. For this reason, it is especially important in the educational context. Personality has an important role because of its relationship to students’ goal orientations, self-esteem, well-being and cognitive level. Personality traits are associated with positive and negative emotions because they determine students’ affective and motivational experiences [17]. Pervin, Cervone, & Oliver [18] define the FFM’s five factors: (1) Neuroticism characterizes the tendency to experience negative emotions. Typical adjectives describing neuroticism are moody, nervous, and touchy. (2) Extraversion characterizes sensation seeking and the quantity and the intensity of interpersonal relationships. Typical adjectives describing extraversion are sociable, assertive, and energetic. (3) Openness to experience characterizes autonomous thinking, a willingness to examine unfamiliar ideas, and an inclination to try new things. Typical adjectives describing openness are inquisitive, philosophical, and innovative. (4) Agreeableness characterizes the quality of interpersonal interactions along a continuum from social antagonism to compassion. Typical adjectives describing agreeableness are kind, considerate, and generous. (5) Lastly, Conscientiousness characterizes a sense of duty, persistence, and self-disciplined goal-directed behavior. Typical adjectives describing conscientiousness are organized, responsible, and efficient.

Zeidner and Matthews [19,20] have established the importance of the Big Five personality model in the sphere of educational psychology. Foundational to this model is its characterization of each of its dimensions [13,21,22]: (1) the hereditary nature of personality traits; (2) validation by consensus: multiple studies have identified the same factors; (3) cross-cultural invariance: research has shown its existence in all cultures; (4) predictive utility: model characteristics have demonstrated their predictive ability in achievement, in laboratory tasks, in vocational interests and on the job. The model establishes five personality characteristics, predictive of other measures of intelligence and of academic achievement:

(1) *Openness* (O). Involves intellectual perception, creativity, curiosity, originality, development of the imagination, foresight, nonconventional thinking, and broad-ranging artistic and intellectual interests. The relationship between openness, learning and scholastic achievement is mediated by other factors like intelligence, the value of the academic materials, learning strategies and activities, expectations of success, and the value of the task. A large body of empirical evidence shows that students high in openness attain better achievement than those low in openness. Notwithstanding, according to other research studies, this variable paradoxically becomes an academic disadvantage in the university context.

(2) *Conscientiousness* (C). Refers to a set of personal attributes needed for learning and academic achievement, such as goal-setting, seeking success, sustained effort, self-control, sense of duty, organization, confidence and efficiency. The exercise of self-discipline and emotional self-regulation is particularly important in learning. A significant body of research has evaluated and confirmed the predictive value of this personal characteristic in academic achievement [23]. Kanfer et al. [24] found this to be the factor most closely linked to regulation of effort, with significatively association between the total complex and the average grade. Research has also shown it to be an independent variable with a clear causal effect on achievement, while also modulated by other factors.

(3) *Extraversion* (E). Students with these characteristics show ease in externalizing; they are sociable, energetic, enthusiastic, adventurous and assertive. Extraverted students prefer other activities to study, and extraversion is found in a moderate, negative correlation to achievement [25]. Introverts have low distraction and better study habits, more ability to stay on task and better consolidation of learning [26]. Extraversion has shown different relations to behavior in primary education, behavior in secondary education and behavior at university, due to the different working environments, more cooperative in the former and more competitive in the latter.

(4) *Agreeableness* (A). These students are characterized by group-orientation, altruism, compliance, warmth, empathy. A positive relationship has been found with academic achievement, in that kindness facilitates cooperation in learning processes [27]. By contrast, antisocial personality traits are associated with the opposite effects.

(5) *Neuroticism* (N). Students with high scores in neuroticism have lower cognitive achievement, because of their anxiety, vulnerability, impulsivity, low self-esteem, low affect and difficulties coping with stressful situations. People that are high in neuroticism (low in emotional stability) tend to focus their attention on their state of emotional tension and frequently experience self-referent cognitions, which interfere with their cognitive performance [19]. Likewise, neuroticism is positively related to low self-efficacy judgments and negatively related to academic achievement. Some meta-analyses have placed this relation between the anxiety trait and measures of academic achievement at about −0.20.

Consequently, personality traits constitute one of the non-intellectual factors that most impacts learning and academic achievement. These factors are nonetheless mediated by multiple variables, such as motivational variables, ability self-concept, test anxiety and achievement motivation, which may have an increasing value in predicting achievement [28]. One recent study has analyzed how personality factors are stable or change over time [5]; stability of scores and change in the mean level at university were examined, both for the general domains of the Big Five (e.g., neuroticism) and the narrower facets that underlie these domains (i.e., self-consciousness, anxiety and depression). Moreover, the study evaluated the longitudinal associations between the Big Five domains and facets, and three aspects of adjustment: self-esteem, academic adjustment and social adjustment at university. The results suggested that the rank order stability of the facets was generally large in all samples, and comparable to that observed for the trait domains. The positive affect facet of Extraversion, and the Conscientiousness facets of achievement striving and reliability, were positively associated with the three adjustment indicators in both samples, while the Neuroticism facets of depression and self-consciousness were negatively associated with adjustment. In summary, the findings show that it can be useful to consider the facets of the Big Five in order to discover the nuances of how personality develops along with adjustment at university.

### 1.3. Big Five Personality Model, Self-Regulation, and Coping Strategies

In an already classic study, Mathews et al. [29] showed how self-regulation is a more interactive and adaptable psychological variable, allowing the more crystallized and static personality construct to materialize under the demands of each situation. “We conceptualize self-regulation as a generic umbrella term for the set of processes and behaviors supporting the pursuit of personal goals within a changing external environment. Self-regulative constructs overlap to a large degree with constructs derived from the transactional theory of stress, such as appraisal and coping [30]”. When they establish the relations between personality and vulnerability to stress, they affirm: “we identify personality traits which relate to adaptive outcome, and self-regulative processes which may mediate these associations…The research area is that of reactions to life stressors. Traits associated with vulnerability to stress may influence appraisal and coping processes”. Research on relations between the two constructions has held the interest of researchers until our day, with an abundance of important contributions [31,32]. Relations between five large personality factors and self-regulation have been hypothesized and confirmed by findings: 

(1) The *Conscientiousness* (C) factor is considered to have a set of attributes belonging to the domain of self-regulation (for example, goal orientation, self-control, industriousness, deliberation, organization and punctuality); it is therefore the factor most clearly associated with different aspects of self-regulation [7]. A meta-analysis has shown a moderate relationship between C and other key factors of motivation, including academic goals, expected outcomes and self-efficacy; a positive relationship was also found with focusing on cognitive tasks, high levels of self-efficacy and low worry [33]. A later study that related the C factor, self-regulated learning and achievement in 52 secondary students concludes that this trait is the most significant predictor of academic achievement [34].

(2) By contrast, *Neuroticism* (N) is associated with several maladaptive behaviors in learning situations, including a low sense of personal control, low self-efficacy as a social agent, and frequently dysfunctional coping strategies, such as avoidance. The N factor has been related to high levels of anxiety, and depletion of cognitive and coping abilities, resulting in reduced learning goals in the face of task demands, and reduced self-regulated performance, processes that require considerable mental resources [35]. Recent research on anxiety has associated deficits in executive processes, including inhibited execution, with deterioration in attentional change processes, leading to a lack of flexibility in deploying one’s attention [36]. Consequently, students high in N would have poorer self-regulation and greater difficulty with coping and adapting to tasks that require activation of effective strategies for overcoming problems in learning situations;

(3) The *Agreeableness* (A) trait relates to social interaction with others. In addition, agreeable students use problem-focused coping strategies related to cooperation [37]. These students can self-regulate in social situations, as part of their competence in individual self-regulation, in cooperative learning situations;

(4) The *Extraversion* (E) trait is especially related to the individual’s confidence when facing settings that have cognitive demands or are socially threatening [38]. Introversion, for its part, has been related to vulnerability caused by stress and certain negative aspects of self-regulation, such as worry and low self-efficacy [29]. Thus, high extraversion is predictive of self-regulated learning;

(5) A limited group of research studies has established a relationship between certain elements of *Openness* (O) and intellectual engagement and self-regulated learning [39]. The relationship between the O factor and learning strategies indicates a connection to one’s engagement in learning [40].

### 1.4. Big Five Personality Model and Achievement Emotions 

The role of personality in being responsible for emotions is important in the educational context [41,42]. Personality traits are positively or negatively associated with educational experience. Extraversion (E) and neuroticism (N) have been considered the two most important traits in relation to achievement emotions. 

The E students have more positive emotions and state of happiness, while introverts depend more on the context and the specific situation. Students high in N have more negative emotional experiences, along with anxiety and frequent anger. Biological models of personality [43] have confirmed these tendencies, demonstrating greater sensitivity to rewards and a greater state of happiness in extraverted persons. By contrast, N people are especially sensitive to punitive stimuli that affect their state of mind [38]. Personality also affects cognitive processes, with greater influence on emotional states. The *E students* tend to appraise their life events in order to change them. Furthermore, they use problem-focused coping strategies, with thoughts and actions for maintaining positive emotional states. By contrast, *N students* perceive and cope with events differently, using self-critical coping strategies that produce negative states. This involves indirect effects on emotions, with interpersonal conflicts and self-inflicted life problems [14].

### 1.5. Aims and Hypotheses

Based on the prior evidence, our research objectives were: (1) to establish a model of linear, empirical, associative relations that validates the relationships between the variables (personality, self-regulation, achievement emotions, and coping) established by previous research, (2) to infer needs and proposals for intervening in emotional health in the different profiles of university students. Specifically, the following hypotheses were posed: 

(1) The personality variables C, O, A, E will have an associative and significant positive, predictive relationship with respect to self-regulation (SR), while N will have an associative and negative predictive relationship with SR.;

(2) The personality variables C, O, A, E and the SR variable will have an associative and significant positive, predictive relationship with respect to positive achievement emotions, and a negative relationship to negative achievement emotions; the opposite will occur in the case of N.;

(3) The personality variables C, O, A, E will have an associative and significant positive, predictive relationship with respect to problem-focused coping strategies, while the N variable will be positively associated with the use of emotion-focused strategies and negatively associated with problem-focused strategies. However, SR will be associated with and negatively predicted by the use of both emotion-focused and problem-focused coping strategies, since SR is associated with fewer experiences of stress;

(4) Emotion-focused coping strategies will positively and significantly predict negative achievement emotions, while they will negatively predict positive achievement emotions. With problem-focused coping strategies, however, the opposite will occur.

## 2. Method

### 2.1. Participants

We used a total sample of 642 undergraduate students from two universities of Spain. The sample was composed of students enrolled in Psychology, Primary Education, and Educational Psychology degree programs; 85.5% were women and 14.5% were men. Their ages ranged from 19 to 25 years, with a mean age of 21.33 years; 324 national students were from one university and the rest from another. An incidental and non-randomized design was used. The participation was anonymous and voluntary, based on an invitation from the teachers (teaching and learning process) of this degree programs in each University. The invitation made to the teachers and students of each university was direct, through the Orientation unit of each University. The completion of the questionnaires in each subject (specific teaching-learning process) was online.

### 2.2. Instruments

*Big Five.* BFQ-N [44], based in Barbaranelli et al. [45] was used. An adaptation for young university students was used [46]. The Confirmatory Analysis (CFA) have reproduced a pentafactorial structure corresponding to the Model of the Big Five. The results have shown adequate psychometric properties and acceptable adjustment rates. The confirmatory model second order showed a good fit [Chi-square = 38.273; Degrees of freedom (20–15) = 5; *p* < 0.001; Normed Fit Index, NFI = 0.939; Relative Fix Index, RFI = 0.917; Incremental Fix Index, IFI = 0.947; Tucker-Lewis Index TLI = 0.937, Comparative Fit Index, CFI = 0.946; Root Mean Square Error of Approximation, RMSEA = 0.065; HOELTER index = 2453 (*p* < 0.05) and, 617 (*p* < 0.01)]. The internal consistency of the total Scale is good (Alpha = 0.956; Part 1 = 0.932, Part 2 = 0.832; Spearman-Brown = 0.962; Guttman = 0.932).

*Self-Regulation.* This variable was measured using the Short Self-Regulation Questionnaire, SSRQ [47]. It has already been validated in Spanish samples [48], and possesses acceptable validity and reliability values, similar to the English version. The Short SRQ is composed of four factors (goal setting-planning, perseverance, decision making and learning from mistakes) and 17 items (all of them with saturations greater than 0.40), with a consistent confirmatory factor structure (Chi-Square = 250.83, *df* = 112, *p* < 0.001; CFI = 0.90, GFI = 0.92, AGFI = 0.90, RMSEA = 0.05). Internal consistency was acceptable for the total of questionnaire items (*α* = 0.86) and for the factors of goal setting-planning (*α* = 0.79), decision making (*α* = 0.72) and learning from mistakes (*α* = 0.72). However, the perseverance factor (*α* = 0.63) showed low internal consistency. Correlations have been studied between each item and its factor total, among the factors, and between each factor and the complete questionnaire, with good results for all, except for the decision-making factor, which had a lower correlation with other factors (range: 0.41–0.58). The correlations between the original version and the complete version, and between the original and the short versions with a Spanish sample (complete SRQ with 32 items and short SRQ with 17 items) are better for the short version (short-original: *r* = 0.85 and short-complete: *r* = 0.94; *p* < 0.01) than for the complete version (complete-original: *r* = 0.79; *p* < 0.01). 

*Coping strategies.* The Escala Estrategias de Coping (Coping Strategies Scale, EEC), was used in its original version [49], as validated for university students [50]. It was constructed according to theoretical-rational criteria, considering the Lazarus and Folkman questionnaire [30] and taking coping assessment studies by Moos and Billings [51] as a basis. Although the original instrument contained 90 items, the validation produced a first-order structure of 64 items and a second order with 10 factors and two dimensions, both of them significant, with adequate fit values in the latter case (Chi-square = 878.750; degrees of freedom (77–34) = 43, *p* < 0.001; NFI = 0.901; RFI = 0.945; IFI = 0.903, TLI = 0.951; CFI = 0.903). Reliability measures are Cronbach alpha of 0.93 (complete scale), 0.93 (first half) and 0.90 (second half), Spearman-Brown of 0.84 and Guttman of 0.80. This questionnaire is composed of 11 factors and two dimensions: (1) *Dimension: emotion-focused coping:* F1. Fantasy distraction; F6. Help for action; F8. Preparing for the worst; F9. Venting and emotional isolation; F11. Resigned acceptance. (2) *Dimension: problem-focused coping:* F2. Help seeking and family counsel; F5. Self-instructions; F10. Positive reappraisal and firmness; F12. Communication of feelings and social support; F13. Seeking alternative reinforcement. 

*Learning Related Emotions*, AEQ [52]. This instrument to include scales for nine different emotions (enjoyment, hope, pride, relief, anger, anxiety, hopelessness, shame, and boredom) was based on two criteria. The AEQ addresses activity emotions (enjoyment, boredom, and anger), prospective outcome emotions (hope, anxiety, and hopelessness), and retrospective outcome emotions (pride, relief, and shame). In terms of valence, the instrument measures both positive and negative emotions, and in terms of activation, it assesses both activating and deactivating emotions. As such, the AEQ makes up the four emotion categories comprising the valence and activation dimensions: positive activating (enjoyment, hope, pride); positive deactivating (relief); negative activating (anger, anxiety, shame); and negative deactivating (hopelessness, boredom). In this sample, the Confirmatory Factorial Analysis (CFA) have reproduced a structure corresponding to the AEQ Model:

(1) *Academic Emotions Class* [53]. The results have shown adequate psychometric properties and acceptable adjustment rates. The confirmatory model showed a good fit [Chi-square = 843.028; Degrees of freedom (44–25) = 19; *p* < 0.001; NFI = 0.954; RFI = 0.967; IFI = 0.953; TLI = 0,958, CFI = 0.971; RMSEA = 0.081; HOELTER=156 (*p* < 0.05) and, 158 (*p* < 0.01)]. The internal consistency of the total Scale is good [Alpha = 0.904; Part 1 = 0.803, Part 2 = 0.853; Spearman-Brown = 0.903 and 0.853; Guttman = 0.862]. An example of items are: item 1: I get excited about goint to class; item 36: I get bored; item 75: I feel so hopeless all my energy is depleted.

(2) *Academic Emotions Study* [54]. The results have shown adequate psychometric properties and acceptable adjustment rates. The confirmatory model showed a good fit [Chi-square = 729,890; Degrees of freedom (44–25) = 19; *p* < 0.001; NFI = 0.964; RFI = 0.957; IFI = 0,973; TLI = 0.978, CFI = 0.971; RMSEA = 0.080; HOELTER = 165 (*p* < 0.05) and, 178 (*p* < 0.01)]. The internal consistency of the total Scale is good (Alpha = 0.939; Part 1 = 0.880, Part 2 = 0.864; Spearman-Brown = 0.913 and 0.884; Guttman = 0.903]). An example of items are: item 90: I get angry when I have to study; item 113: In study sense of confidence motivates me; item 144: I´m proud of myself.

(3) *Academic Emotions Test* [55]. The results have shown adequate psychometric properties and acceptable adjustment rates. The confirmatory model showed a good fit [Chi-square = 376.658; Degrees of freedom (44–25) = 19; *p* < 0.001; NFI = 0,978; RFI = 0.969; IFI = 0.983; TLI = 0.978, CFI = 0.963; RMSEA = 0.080; HOELTER = 169 (*p* < 0.05) and, 188 (*p* < 0.01)]. The internal consistency of the total Scale is good (Alpha = 0.913; Part 1 = 0.870, Part 2 = 0.864; Spearman-Brown = 0.824 and 0.869; Guttman = 0.868). An example of items are: item 170: Before the exam I feel nervous and uneasy. Item 181: I enjoy taking the exam; item 224: I am very satisfied with myself. 

### 2.3. Procedure

Informed consent was obtained from the study participants. The students completed the scales voluntarily using an online platform [56] covering a total of five specific teaching-learning processes, in different university subjects imparted over two academic years. Presage variables was evaluated in September–October of 2018 and of 2019, Process variables in February−March of 2018 and of 2019, and Product variables in May–June of 2018 and of 2019. The procedure was approved by the respective Ethics Committees of the two universities, in the context of R & D Project (2018–2021).

### 2.4. Data Analysis

*Correlation analysis.* For the hypothesis 1 to 3 we are correlated the Personality variables, with Self-regulation, Achievement emotions and Coping strategies variables. As well as the calculation of reliability (Pearson bivariate correlation) through IBM-SPSS Program, v.25 (IBM, New York, NY, USA) [57].

*Confirmatory Factor Analysis and Reliability.* For the hypothesis 4, a Structural Equation Model (SEM) analysis was conducted in this sample. Model fit was assessed by first examining the chi-square to degrees of freedom ratio as well as the Comparative Fit Index (CFI) and Normed Fit Index (NFI), Incremental Fit Index (IFI), and Relative Fit Index (RFI). Ideally, these should be greater than 0.90. We also used the Hoelter Index to determine adequacy of sample size [58]. It´s used the statistical program AMOS Program (v.22) [59].

## 3. Results

### 3.1. Big Five and Self-Regulation

Evidence of associative relations was found in several ways. First, the personality factor conscientiousness (C) had the most significant association with self-regulation (SR), both globally and in its constituent factors (especially in strength of association with perseverance (*r* = 0.600; *p* < 0.001) and goals (*r* = −0.587; *p* < 0.001)), although it was also positively associated with the factors extraversion, agreeableness and openness. Second, the personality factor neuroticism (N) had a significant negative associative relationship with self-regulation (SR), both globally and in its factors; especially important was its negative relation to decision making (*r* = −0.365; *p* < 0.001) and learning from mistakes (*r* = −0.302; *p* < 0.001) (see Table 1).

### 3.2. Big Five and Coping Strategies

A significant positive association appeared between each of the personality factors and total strategies used, although the weakest associative strength corresponded to conscientiousness. The degree of strategy use, however, varied according to the personality factor. While the neuroticism factor was associated positively with total emotion-focused strategies and not associated with total problem-focused strategies, the opposite was true of all remaining factors.

Specific analysis of strategy profiles associated with each personality factor revealed commonalities and differences. The conscientiousness factor was positively associated with all problem-focused coping strategies, especially with the strategies positive reappraisal and firmness (F10) and communicating feelings and social support (F12), and negatively associated with the strategies reducing anxiety and avoidance (F7), preparing for the worst (F8), and emotional venting and isolation (F9). The neuroticism factor, however, had the opposite profile, being positively associated with the former strategies of emotional venting—F9’s high value [*r* = 0.509; *p* < 0.001] warrants special mention, as a health-related risk factor—and negatively associated with the strategies of self-instructions (F5) and positive reappraisal and firmness (F10). 

In complementary fashion, the factors of extraversion, agreeableness and openness to experience showed similar associative patterns. In all cases they were associated with certain *emotion-focused strategies,* either significantly and positively (strategy F6) or significantly and negatively (strategy F9), and above all, with different *problem-focused strategies*, although use of these strategies was somewhat less in the case of openness to experience (See Table 2). 

### 3.3. Big Five and Achievement Emotions

Evidence of important relations was found in the associative relationships. On one hand, a generalized, positive, significant relationship is observed between all the factors of personality and positive emotions, except in the case of neuroticism, which is inversely related, holding a negative association with positive emotions. On the other hand, there is a significant negative relationship between all the personality components and negative emotions, except in the case of neuroticism, which is positively associated with negative emotions. Moreover, the conscientiousness factor was the factor most strongly associated with positive emotions (especially with hope and pride), while neuroticism was the factor with the strongest significant negative association with positive emotions (hope and enjoyment). In complementary fashion, conscientiousness held the strongest negative association with negative emotions (boredom, hopelessness and anger), while neuroticism was the factor with the strongest significant positive association with negative emotions (anxiety, hopelessness and anger). On the other hand, extraversion, agreeableness and openness to experience showed similar relations to those of conscientiousness, in connection with positive and negative emotions. These relationships held constant in the three situations evaluated (class, study time and testing), with a consistent, stable tendency. See Table 3.

### 3.4. Structural Equation Model of Prediction

The results of Structural Equation Model (SEM) showed an acceptable model of relationships, in each emotional context. The relationship parameters of both models are presented below (see Table 4).

In the three models constructed, the Personality (P) variable was a significant positive predictor of Self-regulation (SR) and its components, although differentially by component, with neuroticism being a negative prediction factor (−0.271). It also differentially predicted Coping Strategies: negatively predicting Emotion-Focused Strategies (EF) and positively predicting Problem-Focused Strategies (PF). The same was true of Achievement Emotions (AE), since it positively predicted Positive Emotions (PE) and negatively predicted Negative Emotions (NE). Moreover, the Personality (P) factor positively predicted the components of SR.

For its part, Self-Regulation (SR)—unlike the former factor—negatively predicted the use of Coping Strategies, both emotion-focused (EF) and problem-focused (PF), as well as negatively predicting Negative emotions (NE), but was not a positive predictor of Positive emotions (PE).

Emotion-Focused coping strategies, particularly F9 (emotional venting and isolation) positively predicted negative emotions (NE), but not positive emotions (PE). However, problem-focused coping strategies, with special weight in Factor F2 (Search for help and family advice), did not predict emotions of any type, whether positive or negative.

Finally, positive emotions (PE), with special weight in hope and enjoyment, negatively predicted negative emotions (NE), whose greatest exponents were hopelessness and anger.

A differential effect on these relations appeared in the Tests situation, where the weight of the indices in the relationships increased, showing greater relational strength. Table 5, Table 6 and Table 7 and Figure 1, Figure 2 and Figure 3 shows the total effects of the variables included in the model.

## 4. Discussion

The empirical results have unevenly endorsed the fulfillment of our hypotheses, in the context of relations in a *Top-down model* of personality, coping, and emotion [60]. (1) The first hypothesis—that personality variables C, O, A, E would have an associative and significant positive, predictive relationship with respect to self-regulation (SR), while N would have an associative but negative predictive relationship with SR—was fulfilled in its entirety. All the association effects as well as structural prediction effects found in the three situations have shown that every personality factor, excepting *neuroticism* (low SR), is positively related to SR, albeit not all of them have the same weight or associative strength. As prior research has shown, the Conscientiousness factor had the strongest association with SR (high SR), following by Extraversion, Openness, and Agreeableness (medium SR) [5,34,61].

(2) The second hypothesis—that personality variables C, O, A, E, and SR, would have an associative and significant positive, predictive relationship with respect to positive achievement emotions, and a negative relationship to negative achievement emotions; while the opposite would occur in the case of N—was also fulfilled, though partially. While relationships with the P construct fulfilled the prediction, SR fulfilled only the negative prediction of negative emotions, but not the expected positive prediction of positive emotions. The fact that the SR construct depends linearly on P is probably the reason that its prediction strength is minimized. In any case, the differential effect of the P factors on experiencing positive emotions in the three situations is undeniable, as well as the dampening effect of SR in inhibiting positive emotions. This seems to endorse that the SR construct has a buffering effect on negative emotionality, while having less effect in promoting positive emotionality [62]. Consequently, these aspects should be further investigated in the future.

(3) The third hypothesis projected that personality variables C, O, A, E would have an associative and significant positive, predictive relationship with respect to problem-focused coping strategies, and negative with emotion-focused strategies, while the N variable would be positively associated with the use of emotion-focused strategies and negatively with problem-focused strategies. At the same time, SR would be associated with and negatively predict the use of both emotion-focused and problem-focused coping strategies, since SR is associated with fewer experiences of stress [63]. Both hypotheses were fulfilled. In the first case, the results confirm prior evidence, since personality components were associated with relatively stable coping strategies [64,65,66]. The more harmful emotion-focused strategies were predicted by the N factor, characterized by low self-regulation, while the C factor was usually associated with problem-focused strategies, typical of high self-regulation, and strategies associated with factors E, O, and A were mixed, typical of medium SR [67,68,69]. These tendencies, however, ought to be empirically reviewed, in more precise fashion, before drawing any definite conclusion.

(4) Finally, the fourth hypothesis affirmed that emotion-focused coping strategies would positively and significantly predict negative achievement emotions, while negatively predicting positive achievement emotions; the opposite would occur with problem-focused coping strategies. This hypothesis was only partially fulfilled. In effect, emotion-focused strategies were predictive of negative emotions (hopelessness, anger, anxiety, etc.), but they did not negatively predict positive emotions (enjoyment, hope, pride). Moreover, problem-focused strategies failed to predict any type of emotion. This fact is consistent with the result referring to SR, presented above. SR and problem-focused coping strategies in and of themselves do not predict positive emotionality [70]. These aspects should also be investigated in more depth, in differential fashion, for each BF component and profile.

### Limitations

The present study also has its limitations, which should be addressed in future research. On one hand, the weight of each personality factor in the global weight of the construct has been analyzed, but we have not considered possible combinations or personality typologies using the Big Five model [71]. Nor have we related the possible differential profiles from the model with their associated degree of SR. In other words, could the personality factors be categorized according to their associated degree of SR, in a graded sequence? This possibility should be analyzed in greater depth. It is also vital to make further comparison of clinical and non-clinical samples, in order to establish differences in psychological processes between these two sample types (normalized and psychopathological). Another limitation is that sample was composed of students enrolled in Psychology, Primary Education, and Educational Psychology degree programs; and 85.5% were women and 14.5% were men. Therefore, it would be necessary for future studies to examine if the results are similar in other university studies, as well as in samples with more male participants. Finally, an important limitation is the correlational design used. Indeed, the relationships they observed may reflect other confounding variables. Experimental and longitudinal evidence is needed to make such mechanistic and directional claims.

## 5. Practical Implications

### 5.1. Academic Implications

Research on the role of personality in education can offer great benefits if we incorporate the variables of personality, motivation and affective processes into existing theories [72]. More studies are needed that show the relations between these variables and adaptive behaviors. Future research also needs to show the existence of nonlinear relations between achievement, motivation and mental health. In short, research on personality can contribute elements of interest to educational psychology if it addresses students’ emotional processes [73] and integrates these into the commonly used cognitive and metacognitive models [74]. 

### 5.2. Professional Implications

From a practical point of view, personality assessment can be a critical help to different types of university students. For example, students *high in N* and anxiety and low in self-esteem will need teacher support, receiving positive feedback after successful outcomes, to help them improve their self-esteem; they will also need programs to help with anxiety and stress management [75]. Students *high in E* may need help dealing with distractions and focusing more on their learning goals. Students *low in C* may need help to maintain their interest and application to learning, in other words, an external regulatory context that promotes their self-regulation [76,77,78]. Students *high in O* will need to be involved in routine practices and exercises. Students *high in A* will need to learn to be more assertive with others in the classroom. Finally, teachers can organize activities more flexibly, taking into account students’ affect and performance needs, including goals, learning strategies, and plans.

## 6. Conclusions

As has already been suggested in prior research, self-regulation (as a meta-behavioral construct) may be considered a behavioral materialization of personality factors [21,79]. Something similar also occurs with coping strategies, and hence, with achievement emotions, whether positive or negative. Moreover, this behavior tends to be maintained with a similar pattern in all three situations of the university academic context that we have analyzed here. All this points to the need to place the personality construct in its rightful place in psychoeducational assessment and intervention [80,81] during university academic processes, notwithstanding the consideration that the *person x context* interaction is an essential aspect of understanding the variability in how students experience achievement emotions, as recent research has shown [76,77,82,83] and as the authors of the *Control-Value Theory* have themselves reported [84,85,86,87,88]. 

## Figures and Tables

**Figure 1 ijerph-17-03602-f001:**
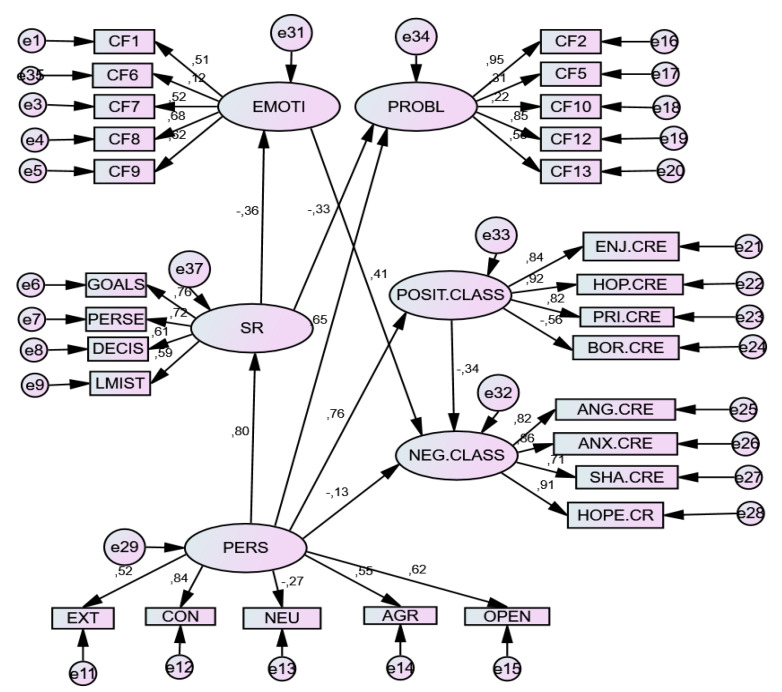
SEM model for Class Situation. Note. E = Extraversion; C = Conscientiousness; N = Neuroticism; A = Agreeableness; O = Openness. *D1. Emotion-focused strategies*: F1. Fantasy distraction; F6. Help for action; F7. Reducing anxiety and avoidance; F8. Preparing for the worst; F9. Emotional venting and isolation; *D2*. *Problem-focused strategies:* F2. Seeking help; F5. Self-instructions; F10. Positive reappraisal and firmness; F12. Communicating feelings and social support; F13. Seeking alternative reinforcement.

**Figure 2 ijerph-17-03602-f002:**
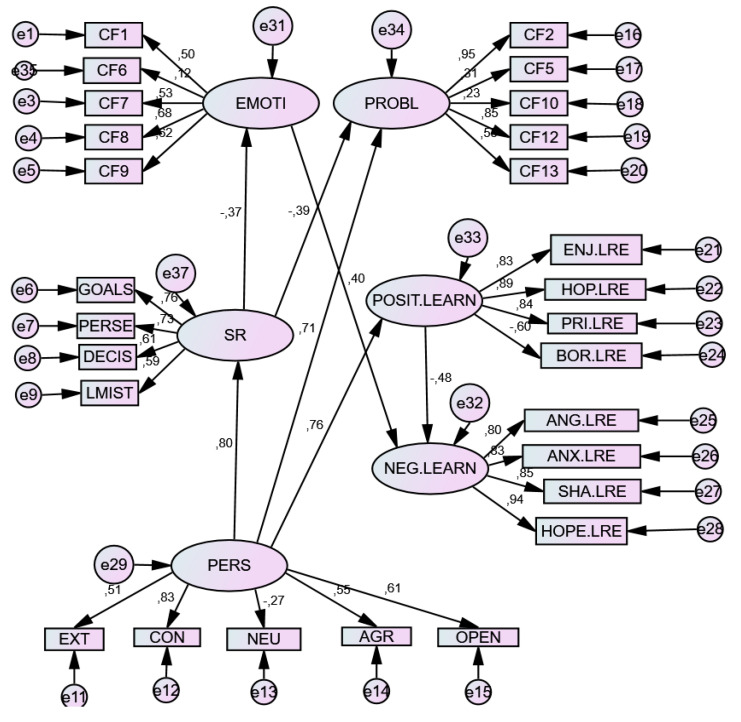
SEM model for Learning Situation. Note. E = Extraversion; C = Conscientiousness; N = Neuroticism; A = Agreeableness; O = Openness. *D1. Emotion-focused strategies*: F1. Fantasy distraction; F6. Help for action; F7. Reducing anxiety and avoidance; F8. Preparing for the worst; F9. Emotional venting and isolation; *D2*. *Problem-focused strategies:* F2. Seeking help; F5. Self-instructions; F10. Positive reappraisal and firmness; F12. Communicating feelings and social support; F13. Seeking alternative reinforcement.

**Figure 3 ijerph-17-03602-f003:**
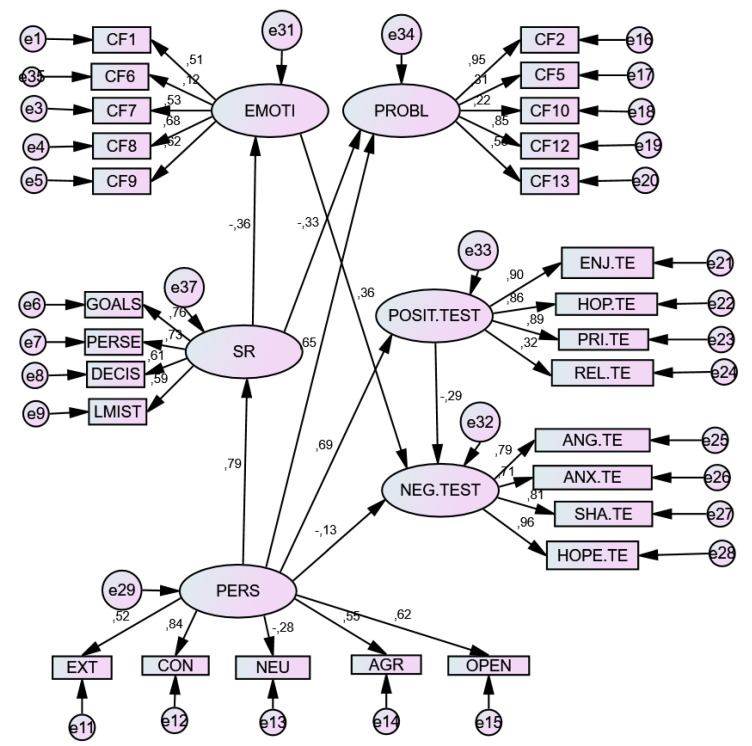
SEM model for Testing Situation. Note. E = Extraversion; C = Conscientiousness; N = Neuroticism; A = Agreeableness; O = Openness. *D1. Emotion-focused strategies*: F1. Fantasy distraction; F6. Help for action; F7. Reducing anxiety and avoidance; F8. Preparing for the worst; F9. Emotional venting and isolation; *D2*. *Problem-focused strategies:* F2. Seeking help; F5. Self-instructions; F10. Positive reappraisal and firmness; F12. Communicating feelings and social support; F13. Seeking alternative reinforcement.

**Table 1 ijerph-17-03602-t001:** Correlations between the Big Five Factors and Self-Regulation (*n* = 642).

Self-Regulation	E	C	N	A	O
SR TOTAL	0.228 **	0.637 **	−0.367 **	0.245 **	0.364 **
GOALS	0.264 **	0.587 **	−0.181 **	0.258 **	0.345 **
PERSEVERANCE	0.109 **	0.600 **	−0.231 **	0.133 **	0.243 **
DECISIONS	0.241 **	0.356 **	−0.365 **	0.135 **	0.282 **
LEAR. MISTAKE	0.222 **	0.383 **	−0.302 **	0.195 **	0.221 **

Note. E = Extraversion; C = Conscientiousness; N = Neuroticism; A = Agreeableness; O = Openness; ** *p* < 0.001.

**Table 2 ijerph-17-03602-t002:** Correlations between *Big Five* factors and *Coping Strategies* (*n* = 642).

Coping Strategies	E	C	N	A	O
TOTAL	0.273 **	0.128 *	0.216 **	0.241 **	0.258 **
*D1. Emotion*	0.002	−0.145 **	0.379 **	0.043	0.074
F1	0.151 **	−0.010	0.213 **	0.032	0.122 **
F6	0.203 **	0.157 **	0.046	0.174 **	0.190 **
F7	0.015 **	−0.137 **	0.144 **	−0.030	−0.020
F8	0.003	−0.152 **	0.409 **	0.009	0.024
F9	−0.197 **	−0.288 **	0.508 **	−0.246 **	0.056
*D2. Problem*	0.407 **	0.332 **	−0.018	0.386 **	0.248 **
F2	0.268 **	0.268 **	0.087	0.316 **	0.063
F5	0.207 **	0.252 **	−0.218 **	0.231 **	0.313 **
F10	0.391 **	0.295 **	−0.295 **	0.299 **	0.416 **
F12	0.318 **	0.270 **	0.118 *	0.330 **	0.070
F13	0.358 **	0.100 **	0.122 *	0.253 **	0.286**

Note. E = Extraversion; C = Conscientiousness; N = Neuroticism; A = Agreeableness; O = Openness; *D1. Emotion-focused strategies*: F1. Fantasy distraction; F6. Help for action; F7. Reducing anxiety and avoidance; F8. Preparing for the worst; F9. Emotional venting and isolation; *D2*. *Problem-focused strategies:* F2. Seeking help; F5. Self-instructions; F10. Positive reappraisal and firmness; F12. Communicating feelings and social support; F13. Seeking alternative reinforcement; **p* < 0.01; ** *p* < 0.001.

**Table 3 ijerph-17-03602-t003:** Correlations between Big Five factors and Achievement Emotions (*n* = 642).

Situations	E	C	N	A	O
**Class**					
*Positives*	0.390 **	0.586 **	−0.143 *	0.325 **	0.523 **
Enjoyment	0.280 **	0.482 **	−0.152 **	0.275 **	0.443 **
Hope	0.382 **	0.689 **	−0.204 **	0.341 **	0.517 **
Pride	0.369 **	0.515 **	−0.066 *	0.277 **	0.413 **
*Negatives*	−0.297 **	−0.362 **	0.378 **	−0.228 **	−0.338 **
Boredom	−0.081 *	−0.437 **	0.335 *	−0.193 **	−0.202 **
Anger	−0.177 *	−0.384 **	0.340 **	−0.304 **	−0.260 **
Anxiety	−0.223 **	−0.272 **	0.418 **	−0.160 **	−0.332 **
Shame	−0.312 **	−0.199 **	0.338 **	−0.142 **	−0.290 **
Hopelessness	−0.172 *	−0.400 **	0.374 *	−0.224 **	−0.354 **
**Study**					
*Positives*	0.324 **	0.577 **	−0.161 **	0.352 **	0.478 **
Enjoyment	0.292 **	0.526 **	−0.112 **	0.291 **	0.498 **
Hope	0.357 **	0.528 **	−0.225 **	0.306 **	0.406 **
Pride	0.303 **	0.505 **	−0.079 *	0.277 **	0.397 **
*Negatives*	−0.219 **	−0.341 **	0.446 **	−0.195 **	−0.354 **
Boredom	−0.153 **	−0.515 **	0.339 **	−0.251 **	−0.296 **
Anger	−0.068	−0.340 **	0.443 **	−0.213 **	−0.233 **
Anxiety	−0.162 **	−0.255 **	0.438 **	−0.110 **	−0.274 **
Shame	−0.251 **	−0.254 **	0.412 **	−0.170 **	−0.290 **
Hopelessness	−0.231 **	−0.348 **	0.441 **	−0.215 *	−0.374 **
**Test**					
*Positives*	0.352 **	0.534 **	−0.176 **	0.245 **	0.480 **
Enjoyment	0.292 **	0.485 **	−0.110 **	0.200 **	0.435 **
Hope	0.327 **	0.525 **	−0.269 **	0.268 **	0.469 **
Pride	0.341 **	0.498 **	−0.137 **	0.265 **	0.411 **
*Negatives*	−0.129 **	−0.295 **	0.487 **	−0.179 **	−0.319 **
Relief	0.099 **	0.215 **	0.086 **	0.168 **	0.120 **
Anger	−0.061	−0.258 **	0.415 **	−0.205 **	−0.257 **
Anxiety	−0.070	−0.108 **	0.438 **	−0.025	−0.420 **
Shame	−0.142 **	−0.217 **	0.394 **	−0.133 **	−0.197 **
Hopelessness	−0.191 **	−0.353 **	0.405 **	−0.212 **	−0.348 **

Note. E = Extraversion; C = Conscientiousness; N = Neuroticism; A = Agreeableness; O = Openness; * *p* < 0.01; ** *p* < 0.001.

**Table 4 ijerph-17-03602-t004:** Models of structural linear results of the variables.

Model	DF	Chi-Square	*p-Value*	NFI	RFI	IFI	TLI	CFI	RMSEA	Hoelter 0.05–0.01
1. Class	(405–89): 316	3725,454	0.001	0.953	0.960	0.951	0.900	0.900	0.080	247–260
2. Study	(434–92): 114	3800,460	0.001	0.903	0.932	0.954	0.908	0.900	0.080	239–257
3. Test	(405–89): 316	3358,313	0.001	0.901	0.917	0.916	0.901	0.900	0.080	228–245

Note. Model in each Emotional Context (1 to 3).

**Table 5 ijerph-17-03602-t005:** Standardized Total Effects (Default model): Class situation.

CLASS	P	SR	EF	PF	PE	NE
SELF-REGUL	0.792					
EMOT-FOCUS	−0.283	−0.357				
PROB-FOCUS	−0.387	−0.328				
POSIT-EMOT	0.759					
NEGAT-EMOT	−0.520	−0.144	0.404		−0.242	
CONSCIENT	0.836					
OPEN EXPER	0.622					
AGREEABLEN	0.550					
EXTRAVERS	0.518					
NEUROTI	−0.217					
GOALS	0.601	0.758				
PERSEVER	0.574	0.725				
DECISIONS	0.482	0.609				
LEARN. MIST	0.467	0.590				
CF8	−0.191	−0.242	0.678			
CF9	−0.176	−0.222	0.621			
CF7	−0.148	−0.187	0.525			
CF1	−0.143	−0.181	0.506			
CF6	−0.034	−0.043	0.120			
CF2	0.368	−0.312		0.935		
CF12	0.329	−0.278		0.845		
CF13	0.211	−0.129		0.545		
CF5	0.121	−0.110		0.314		
CF10	0.087	−0.078		0.225		
HOPE	0.710				0.935	
ENJOY	0.629				0.829	
PRIDE	0.629				0.828	
HOPELESSNESS	−0.478	−0.132	0.371		−0.274	0.918
ANGER	−0.450	−0.125	0.324		−0.258	0.865
ANXIETY	−0.423	−0.117	0.328		−0.242	0.813
BOREDOM	−0.389	−0.108	0.302		−0.274	0.749
SHAME	−0.343	−0.095	0.268		−0.194	0.660

Note. D1. *Emotion-focused strategies*: F1. Fantasy distraction; F6: Help for action; F7. Reducing anxiety and avoidance; F8: Preparing for the worst; F9. Emotional venting and isolation; D2. *Problem-focused strategies*: F2. Seeking help; F5. Self-instructions; F10. Positive reappraisal and firmness; F12. Communicating feelings and social support; F13. Seeking alternative reinforcement.

**Table 6 ijerph-17-03602-t006:** Standardized Total Effects (Default model): Study Situation.

STUDY	P	SR	EF	PF	PE	NE
SELF-REGUL	0.801					
EMOT-FOCUS	−0.360	−0.369				
PROB-FOCUS	0.405	−0.449				
POSIT-EMOT	0.760					
NEGAT-EMOT	−0.562	−0.227	0.354			
CONSCIENT	0.838					
OPEN EXPER	0.622					
AGRAD	0.548					
EXTRAVERS	0.516					
NEUROTI	−0.290					
GOALS	0.600	0.756				
PERSEVER	0.574	0.716				
DECISIONS	0.495	0.618				
LEARN. MIST	0.474	0.591				
CF8	−0.232	−0.212	0.681			
CF9	−0.234	−0.292	0.649			
CF7	−0.182	−0.227	0.507			
CF1	−0.162	−0.203	0.451			
CF6	−0.134	−0.123	0.131			
CF2	0.381	−0.347		0.942		
CF12	0.347	−0.206		0.856		
CF13	0.226	−0.152		0.559		
CF5	0.128	−0.116		0.315		
CF10	0.091	0.083		0.228		
HOPE	0.660				0.865	
ENJOY	0.652				0.868	
PRIDE	0.658				0.858	
HOPELESSNESS	−0.514	−0.208	0.169			0.915
ANGER	−0.467	−0.191	0.155			0.839
ANXIETY	−0.472	−0.189	0.154			0.831
BOREDOM	−0.436	−0.176	0.155			0.775
SHAME	−0.467	−0.189	0.154			0.831

Note. D1. *Emotion-focused strategies*: CF1. Fantasy distraction; CF6: Help for action; CF7. Reducing anxiety and avoidance; CF8: Preparing for the worst; CF9. Emotional venting and isolation; D2. *Problem-focused strategies:* CF2. Seeking help; CF5. Self-instructions; CF10. Positive reappraisal and firmness; CF12. Communicating feelings and social support; CF13. Seeking alternative reinforcement.

**Table 7 ijerph-17-03602-t007:** Standardized Total Effects (Default model): Exam situation.

EXAM	P	SR	EF	PF	PE	NE
SELF-REGUL	0.793					
EMOT-FOCUS	−0.283	−0.357				
PROB-FOCUS	0.383	−0.333				
POSIT-EMOT	0.694					
NEGAT-EMOT	−0.430	−0.130	0.365		−0.288	
CONSCIENT	0.838					
OPEN EXPER	0.616					
AGRAD	0.548					
EXTRAVERS	0.516					
NEUROTI	−0.277					
GOALS	0.600	0.757				
PERSEVER	0.576	0.727				
DECISIONS	0.486	0.610				
LEARN. MIST	0.465	0.586				
CF8	−0.192	−0.242	0.677			
CF9	−0.174	−0.220	0.616			
CF7	−0.151	−0.191	0.533			
CF1	−0.144	−0.182	0.508			
CF6	0.033	−0.042	0.116			
CF2	0.371	−0.317		0.953		
CF12	0.331	−0.283		0.849		
CF13	0.212	−0.182		0.545		
CF5	0.122	−0.104		0.314		
CF10	0.087	−0.075		0.225		
HOPE	0.624				0.900	
ENJOY	0.617				0.890	
PRIDE	0.624				0.862	
HOPELESSNESS	−0.413	−0.125	0.350		−0.227	0.960
ANGER	−0.349	−0.103	0.287		−0.234	0.810
ANXIETY	−0.339	−0.106	0.295		−0.227	0.788
BOREDOM	−0.307	−0.093	0.260		−0.206	0.713
SHAME	−0.223				0.327	

Note. D1. *Emotion-focused strategies*: CF1. Fantasy distraction; CF6: Help for action; CF7. Reducing anxiety and avoidance; CF8: Preparing for the worst; CF9. Emotional venting and isolation; D2. *Problem-focused strategies:* CF2. Seeking help; CF5. Self-instructions; CF10. Positive reappraisal and firmness; CF12. Communicating feelings and social support; F13. Seeking alternative reinforcement.

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
