# Peer review of "Big Five, Self-Regulation, and Coping Strategies as Predictors of Achievement Emotions in Undergraduate Students"

_ijerph, 2020, doi:10.3390/ijerph17103602_

Round 1

Reviewer 1 Report

This is an interesting paper on the analysis of linear relations between personality, self-regulation, coping strategies and achievement emotions in undergraduate students.

The authors present a good paper, well-structured, and with consistent results. The literature review is updated and the discussion describes the results of the study and compares with the literature. It is a good study because provides a new perspective on relations between personality, self-regulation, coping strategies and achievement emotions.

Although the manuscript is interesting, there are some minor issues, which can improve the quality of the paper:

-The authors indicate the significance of the results in different ways. For example.

  1. Probability level = ,000; 
  2. (p<.05) and (p<.01). 

Please unify it according to APA style.

-The sample was composed of students enrolled in Psychology, Primary Education, and Educational Psychology degree programs; and 85.5% were women and 14.5% were men. 

Therefore, it would be necessary for future studies to examine if the results are similar in other university studies, as well as in samples with more male participants.

-The authors do not acknowledge the limitation of their correlational design. Indeed, the relationships they observed may reflect other confounding variables. Experimental and longitudinal evidence is needed to make such mechanistic and directional claims. 

Author Response

Dear reviewer: 

Thank you for yours sugestions.

This is an interesting paper on the analysis of linear relations between personality, self-regulation, coping strategies and achievement emotions in undergraduate students.

1) The authors present a good paper, well-structured, and with consistent results. The literature review is updated and the discussion describes the results of the study and compares with the literature. It is a good study because provides a new perspective on relations between personality, self-regulation, coping strategies and achievement emotions.

RESPONSE: Thank you

2) Although the manuscript is interesting, there are some minor issues, which can improve the quality of the paper:

-The authors indicate the significance of the results in different ways. For example.

  1. Probability level = ,000; 
  2. (p<.05) and (p<.01). 

Please unify it according to APA style.

Response: All the indices have been reviewed in the complete document.

3) The sample was composed of students enrolled in Psychology, Primary Education, and Educational Psychology degree programs; and 85.5% were women and 14.5% were men. 

Therefore, it would be necessary for future studies to examine if the results are similar in other university studies, as well as in samples with more male participants.

 Response: This limitation has been inserted and as a proposal for future research

4) The authors do not acknowledge the limitation of their correlational design. Indeed, the relationships they observed may reflect other confounding variables. Experimental and longitudinal evidence is needed to make such mechanistic and directional claims. 

Response: This limitation has been inserted and as a proposal for future research

Reviewer 2 Report

A few thoughts:
-deepening the composition of the sample (how many students in all are there in the two universities? Are there international students?...);
-the fact that the sample is not random but self-selected.
-Line 402: SEM does not match pathway analysis;

-no need to report the standard deviation of age,

-some words to correct (for instance, line 305 sudent);

-standardize bibliography (& instead of and).

Author Response

Dear reviewer:

Thank you for yours sugestions.

A few thoughts:

1) -deepening the composition of the sample (how many students in all are there in the two universities? Are there international students?...);

-the fact that the sample is not random but self-selected.

Response: this information has been inserted

2)

-Line 402: SEM does not match pathway analysis;

-no need to report the standard deviation of age,

-some words to correct (for instance, line 305 sudent);

-standardize bibliography (& instead of and).

Response: everything has been revised and adjusted to the editorial format